EFNet: estimation of left ventricular ejection fraction from cardiac ultrasound videos using deep learning

Ali Waqas 1 waqas.ali2@uet.edu.pk
Alsabban Wesam 2
Shahbaz Muhammad 1
Al-Laith Ali 3
Almogadwy Bassam 4
1 Computer Science Department, University of Engineering and Technology , Lahore , Pakistan
2 Department of Computer and Network Engineering, College of Computing, Umm Al-Qura University , Makkah , Saudi Arabia
3 Computer Science Department, University of Copenhagen , Copenhagen , Denmark
4 Department of Artificial Intelligence and Data Science, Taibah University , Medina , Saudi Arabia
Coelho Paulo Jorge
Electronic publication date: 2025 Jan 21
Publication date: 2025
Volume: 11
Electronic Location ID: e2506
Received 2024 Jul 10; Accepted 2024 Oct 21
Copyright: © 2025 Ali et al.
Copyright year: 2025
Copyright holder: Ali et al.
License: This is an open access article distributed under the terms of the Creative Commons Attribution License, which permits unrestricted use, distribution, reproduction and adaptation in any medium and for any purpose provided that it is properly attributed. For attribution, the original author(s), title, publication source (PeerJ Computer Science) and either DOI or URL of the article must be cited.
License URL: https://creativecommons.org/licenses/by/4.0/

Keywords: Medical imaging, Echocardiography, CNN, Transformers, Heart disease

Funding: The authors received no funding for this work.

==============================
The ejection fraction (EF) is a vital metric for assessing cardiovascular function through cardiac ultrasound. Manual evaluation is time-consuming and exhibits high variability among observers. Deep-learning techniques offer precise and autonomous EF predictions, yet these methods often lack explainability. Accurate heart failure prediction using cardiac ultrasound is challenging due to operator dependency and inconsistent video quality, resulting in significant interobserver variability. To address this, we developed a method integrating convolutional neural networks (CNN) and transformer models for direct EF estimation from ultrasound video scans. This article introduces a Residual Transformer Module (RTM) that extends a 3D ResNet-based network to analyze (2D + t) spatiotemporal cardiac ultrasound video scans. The proposed method, EFNet, utilizes cardiac ultrasound video images for end-to-end EF value prediction. Performance evaluation on the EchoNet-Dynamic dataset yielded a mean absolute error (MAE) of 3.7 and an R2 score of 0.82. Experimental results demonstrate that EFNet outperforms state-of-the-art techniques, providing accurate EF predictions.

Introduction

The diagnosis and treatment of numerous cardiovascular illnesses relies heavily on cardiac ultrasound imaging. Using high-frequency sound waves, cardiac ultrasound, sometimes referred to as echocardiography, is a noninvasive diagnostic method for visualizing the structure and operation of the heart (Lindner, 2009; Čelutkienė et al., 2020). Cardiac ultrasonography is a crucial imaging technique in medicine because it provides helpful data for the diagnosis and treatment of a variety of cardiovascular conditions (Faust et al., 2017). It enables the evaluation of heart function, including ejection fraction (EF), as well as the identification of anatomical anomalies such as valve stenosis or regurgitation.

Transformers in deep learning have emerged as rival architectures for convolutional neural networks (CNNs) and have exhibited comparable efficacy in many computer vision tasks. Deep learning techniques employing both transformers and CNNs have achieved state-of-the-art performance in automated medical image segmentation (Khan, Lee & Lee, 2023). Various imaging modalities have been employed for real-time clinical evaluation and visualization in medical imaging. For example, echocardiography produces spatiotemporal data comprising sequences of 2D images (Bosco et al., 2023). In the context of spatiotemporal data, it is necessary to consider temporal and spatial information in sequences for accurate diagnosis. To detect anomalies and particular diseases, cardiologists are interested in considering the temporal factor information during the measurement of left ventricular ejection fraction (LVEF) or heart wall health analysis (Lara Hernandez et al., 2021). LVEF can be determined by analyzing the apical two-chamber (a2c) or apical four-chamber (a4c) images of the heart and is calculated as the ratio between stroke volume, that is, the difference between end-diastolic volume (EDV) and end-systole volume (EVD), and the ventricle blood volume at end-diastole (Nosir et al., 1997). Heart failure (HF), a dangerous condition in which the heart is unable to pump sufficient blood, can be predicted by measuring the LVEF (Hirata et al., 2009). If cardiologists identify heart failure early through accurate estimation of LVEF, they can start the treatment earlier, and a better prognosis may be achieved (Wang et al., 2015). Heart failure is usually identified by LVEF below the normal range (52–72%) in male and (54–74%) (Lang Roberto et al., 2015). Echocardiography can be used to determine the LVEF, wall thickness, and volume of the left ventricle. However, there are a number of factors that can impact the quality of ultrasound results, including operator skillfulness, noise, aberrations, and diminished contrast, all of which may contribute to the reproducibilty of diagnosis among patients and for the same patient (Voorhees & Han, 2015).

The efficiency of heart blood pumping, as measured by EF, is a basic feature and crucial sign of cardiac health. In individuals with heart disease, low EF is associated with a higher risk of mortality and morbidity (Solomon et al., 2005). Therefore, accurate EF measurement is crucial for the proper diagnosis, planning of the course of therapy, and monitoring of patients with cardiac dysfunction. Owing to its restricted ability to visualize the intricate three-dimensional (3D) anatomy of the heart, traditional two-dimensional (2D) echocardiography is used herein to determine EF. Recent technological advancements have resulted in the development of 3D echocardiography, which offers more specific details on cardiac anatomy and function. However, precise measurement of EF using 3D echocardiography remains challenging owing to the complexity of the data and the need for accurate segmentation of cardiac chambers. Recently, automated methods have been proposed for segmenting and measuring EF from cardiac ultrasound videos. In this article, we suggest a novel method for predicting EF in cardiac ultrasound videos by exploring 3D residual and transformer modules in 2D echocardiography data.

Transformers leverage multi-head self-attention (MHSA) units to comprehend the whole interdependence among the components of an input sequence. Unfortunately, transformers require a large amount of training data and additional training time because of their high computational complexity. In this study, we used transformers to directly estimate the EF from ultrasound footage. With the help of a Residual Transformer Module (RTM) that we name EFNet, we extend the 3D ResNet-based network (Tran et al., 2018) to design an improved EF-estimator.

By using convolutional layers and residual connections, RTM enables both local and global feature representations. The RTM’s global self-attention mechanism helps improve this representation. In the study of 2D + t spatiotemporal ultrasound video images and the direct prediction of EF, EFNet represents a neural network that successfully fuses transformers and CNNs.

Our study focused on the effectiveness of 3D residual and transformer modules in predicting EF in cardiac ultrasound videos, the impact of different training strategies on performance, and whether combining these modules can improve accuracy. We present an end-to-end approach for estimating LVEF by using only ultrasound video images. Our approach incorporates a unique residual transformer module that includes temporal position encoding into a 3D ResNet-based neural network (3D MHSA). To the best of our knowledge, EFNet is an important framework for automatically estimating EF using trained and verified cardiac ultrasound video data. This contribution presents significant implications for the field of cardiology and provides a novel approach for predicting LVEF with the potential for clinical adoption.

The remainder of this article is organized as follows. First, the state-of-the-art contributions for ejection fraction estimation using deep learning methods are reviewed in “Introduction”. “Related Work” describes the proposed method and the inner workings of the proposed modular architecture. “Methodology” presents the experiments and results as well as the implementation aspects of the proposed architecture. “Performance and Discussion” presents the evaluation metrics, a performance comparison with state-of-the-art algorithms, and an ablation study. “Evaluation Metrics” discusses the achieved results and the factors that helped achieve improved performance. The article concludes by summarizing the newly obtained key results and outlining future directions to further extend the introduced line of research.

Related work

In recent years, methods based on deep learning have been used to autonomously determine EF from cardiac ultrasound pictures and videos. For instance, a 3D CNN for EF prediction from 3D echocardiography, and on a dataset of 150 patients it produced highly accurate results (Jafari et al., 2019). Similar to this approach, a multitask 3D CNN for concurrent LV segmentation and EF prediction from 3D echocardiography, obtaining high accuracy on a dataset of 400 patients (Ahn et al., 2021). Several deep learning architectures have been suggested to increase the ability of 3D echocardiography to segment the left ventricle more accurately. To segment the left ventricle (LV) using 3D echocardiography, Vesal, Ravikumar & Maier (2019) suggested a 3D fully convolutional neural network with dilated convolutions, achieving state-of-the-art results in the MICCAI 2017 LV segmentation challenge.

Additional studies have been reported in the literature to more accurately segment the left ventricle and measure EF (Adekkanattu et al., 2023; Ahmad et al., 2022; Bachtiger et al., 2022). The approach (EchoCoTr) suggested in Muhtaseb & Yaqub (2022) considers the challenge of calculating LVEF using ultrasound movies by leveraging the power of vision transformers and CNNs. This method mapped on the EchoNet dynamic state-of-the-art dataset yielded an MAE of 3.92 and R2 score of 0.82. Bi-LSTM layers were used in the sequential model architecture of Wang et al. (2021), and heart failure was predicted by analyzing the reduced EF. Silva et al. (2018) estimated the EF from transthoracic echocardiography (TTE) tests using an advanced 3D CNN equipped with a residual learning block. To forecast heart failure with reduced EF (HFrEF), Ouyang et al. (2020) proposed a deep learning approach that integrates the results of semantic segmentation with clip-level EF prediction using a spatiotemporal CNN. Exploiting similar features, Li et al. (2022) combined the advantages of spatiotemporal transformers with 3D CNNs. The MCLAS framework, which stands for multi-task semi-supervised, allows for accurate EF estimation using echocardiographic sequences from two cardiac perspectives, and was proposed in Wei et al. (2023). A segmentation model for the left ventricle that combines a CNN and transformer architectures was developed in Shi et al. (2022).

In general, echocardiographic characteristics are first extracted using a transformer and CNN architectures, and then the fusion module combines the image features. To create a segmentation map, the attention weight was assessed using a bridge attention network in accordance with the three-layer fusion characteristics. Thomas & Gilbert (2022) employed graph convolutional networks for segmenting left ventricle key points and computed EF using segmentation volumes at the ES and ED frames, which were identified manually. An MAE score of 4.01 was reported for a 16-frame video (only frames between ES and ED) for EchoGraphs with a regression head. For the entire video, an MAE of 4.05 was reported. Reynaud et al. (2021) employed a transformer comprising a BERT model and a residual auto-encoder network to predict the ED and ES frames along with EF estimation, and reported an MAE score of 5.95. EchoGNN, a novel model that utilizes graph neural networks (GNNs), was proposed by Mokhtari et al. (2022) to estimate EF from echocardiogram videos. EchoGNN first obtains a latent echo graph from a single frame or series of frames in an echo cine sequence. EchoGNN then infers the weights of the graph’s nodes and edges, reflecting the importance of specific frames in aiding the EF estimate. Finally, the GNN regressor employs a weighted graph to forecast EF. EchoGNN has an MAE score of 4.45. Following a different route, Gu et al. (2021) employed many attention layers in a CNN architecture and proposed a thorough attention-based CNN (CA-Net) for more precise and explicable medical image segmentation that is simultaneously aware of the most crucial spatial locations, channels, and scales. EchoNet-Peds (Reddy et al., 2023) makes it possible to automate EF measurement accurately and quickly and to identify systolic dysfunction in a pediatric population, both of which have the potential to improve diagnostic ability. The multitask deep learning network echoEFNet developed in Li et al. (2023) consists of a backbone employing dilated convolution, a left ventricular segmentation branch, and a landmark detection branch.

In terms of the differentiating contribution of this article relative to existing articles, as per the experimental findings, the proposed method EFNet surpasses state-of-the-art cutting-edge techniques and accurately predicts the EF.

Methodology

A high-level interpretation of the proposed approach for cardiac ejection fraction prediction (CEFP) is shown in Fig. 1. To extract high-level features from the echocardiography videos, we employed 3D ResNet-18. The residual transformer module is equipped with a 3D (MHSA) module and convolutional layers to model the global and local feature representations. RTMs are used in place of ResNet residual modules in the 3rd and 5th layers in our 3D ResNet-18.

Figure 1 Architecture.

Overview of the EFNet approach for estimating EF using US video scans. Two RTMs are used in EFNet to modify two of the 3D ResNet-18’s residue left modules with the 3D MHSA’s relative positional encoding (RPE). RPE is determined as the sum of the position encodings for height (Rh), width (Rw), and time (Rt). Only one attention head is shown for clarity. The network performs a single-segment prediction using 16 consecutive frames as its input. Once all frames for a patient are divided into 16-frame segments without any overlap, an average of all segment forecasts for that patient is created.

Dataset and preprocessing steps

Medical researchers exploring heart function have greatly benefited from exploring the Echonet-Dynamic dataset (Ouyang et al., 2019). This dataset was also used in this study to assess the connection between left ventricular volumes and EF (Blaivas & Blaivas, 2022). In total, 10,036 movies exhibiting a four-chamber view of the heart were included in the dataset, with a representation of males (52%) and females (48%). The average age of each participants is 68 years, with a standard deviation of 21 years. The key echocardiographic parameters are highlighted in Table 1, including an average ejection fraction of 55.7% (SD 12.5), end systolic volume of 43.3 mL (SD 34.5), and end diastolic volume of 91.0 mL (SD 45.7). The imaging parameters, such as frames per second (50.9, SD 6.8) and the number of frames (175, SD 57), indicate high temporal resolution, supporting the dynamic analysis of cardiac function. These statistics offer insights into the dataset’s robustness and its potential for generalizable cardiac research applications. These recordings represent a wide and representative sample because they range in length, frame rate, and visual quality. Each frame had 112 × 112 pixels, offering a high degree of detail for the examination. It is feasible to evaluate the heart activity over time because at least one film shows a whole cardiac cycle, and other recordings include three to four cycles. The end-systolic (ES) and end-diastolic (ED) volumes, measured in milliliters (mL), represent the main focus of this investigation. During one cycle of each video, we accessed the ED and ES frames with the assistance of medical doctors. Medical doctors of University of Engineering and Technology Health Clinic were involved for manually annotated the ES and ED frames based on the cardiac cycle understanding criteria. ED is identified when the left ventricle is at its largest volume, typically just before the mitral valve closes, corresponding with the onset of the QRS complex on the ECG. ES is marked when the left ventricle is at its smallest volume after contraction, occurring just before the mitral valve reopens, usually correlating with the end of the T wave on the ECG.

Table 1 Dataset statistics (Ouyang et al., 2019).

Statistic	Value	
Number of videos	10,036	
Female (%)	4,885 (48%)	
Age (Years)	68 (21)	
Frames per second	50.9 (6.8)	
Number of frames	175 (57)	
Ejection fraction (%)	55.7 (12.5)	
End systolic volume (mL)	43.3 (34.5)	
End diastolic volume (mL)	91.0 (45.7)	

Table 2 presents the annotated ES and ED frames for the first five videos, with the remaining data represented by ellipses.

Table 2 Annotated ES and ED frames for the first five videos, with the rest represented by ellipses.

Video ID	ED frame	ES frame	
0X100E3B8D3280BEC5	V1_ED	V1_ES	
0X100E491B3CD58DE2	V2_ED	V2_ES	
0X100F044876B98F90	V3_ED	V3_ES	
0X1002E8FBACD08477	V4_ED	V4_ES	
0X1005D03EED19C65B	V5_ED	V5_ES	
…	

To conduct and assess the performance of the study, the dataset is divided into two groups, with 80% of the videos to be utilized for training and the remaining 20% for testing and validation.

We illustrate the cardiac cycle by inserting frames of end-systole and end-diastole from the ultrasound videos, as shown in Fig. 2. The use of these frames helps in better understanding the cardiac cycle and the role of EF in assessing cardiac function. The figure consists of two rows of images: the first row depicts the end-diastolic phase of five different videos, while the second row shows the end-systolic phase of the same five videos. The ED and ES phases are commonly used to describe different stages of the cardiac cycle and are important markers for understanding heart function and disease. We read echocardiography video frame chunks from the EchoNet Dynamic dataset. A few data augmentation techniques are used, including rotation (between −25 and 25 degrees), horizontal flipping (50% probability), random brightness contrast, picture compression (10% likelihood), and blurring (50% probability). We kept the frames at their original size of 112×112. EF is calculated using the following formula:

(1) EF=(EDV−ESV)EDV×100.

Figure 2 ED and ES frames from some videos taken from the EchoNet-Dynamic.

We used the following formula to normalize all the pixels on video frames:

(2) image=(image−mean×max_pixel_value)(std×max_pixel_value)

where mean = std = 0.5 (grayscale images). After completing all pre-processing procedures, the video was passed as input to our EFNet model.

Feature extraction

In EFNet, in order to produce high-level (2D + t) spatiotemporal feature interpretation, we utilize the 3D ResNet-18 (Tran et al., 2018) as the base network. The network receives an echocardiogram video sequence SUS∈RT0×1×H0×W0 as its initial input with frame number T0, height H0, and width W0. We used RTM at two levels in 3D ResNet-18 at layers 2 and 4. After layer one, the level feature map is passed to RTM where T1=T0, H1=H02, and W1=W02. After the second layer, the low-level feature map is passed to RTM where T2=T02, H2=H04, and W2=W04. It is converted to a series of resolution feature maps S′US∈RT1×D1×H1×W1 using convolutional residual modules, where T1=T04, D1=512, H1=H08, and W1=W08. The RTM is provided with multi-channel, low-resolution feature grid sequences.

After the fourth layer, the final low-level feature map is passed to RTM, where T2=T04, H2=H08, and W2=W08. It is converted to a series of resolution feature maps S′US∈RT1×D1×H1×W1 using convolutional residual modules, where T1=T08, D1=512, H1=H016, and W1=W016. The RTM is fed with multi-channel, low-resolution feature grid sequences.

Residual transformer module

Convolution layers, ReLUs, and batch normalization serve as the foundation for residual modules. With the aid of skip connections, the input is transferred to the output of the levels that came before it, and this structure is repeated several times (He et al., 2016). By incorporating temporal position encoding into MHSA, our residual transformer module (RTM) is expanded to the 3D space. In the residual bottleneck module, we employ MHSA rather than 3×3 convolution to lower the computing cost of the more complex ResNet designs. 3D ResNets frequently present shallower depths and lack Bottleneck blocks. We have replaced the third and the last convolutional layer with MHSA in the residual module in order to leverage the self-attention mechanism in shorter ResNets. As a result, RTM is modeled as

(3) y=BN(MHSA(σ(BN(Conv(x)))+x))

where BN denotes the batch normalization, Conv represents the convolutional layer, σ stands for ReLU, and x and y are the RTM’s input and output vectors, respectively.

3D multi-head self attention

In order to learn various attention interpretations at diverse locations through cooperative training, several self-attention heads must be generated as opposed to one, and the results are then combined (Shazeer et al., 2017). Given that this operation is permutation-invariant, positional information must be included by adding positional encoding r. Absolute (like sinusoidal) or relative positional encodings (RPE) might be utilized depending on the purpose. We include temporal positional encoding into the 2D RPE in order to analyze 2D + t ultrasound films with MHSA. Positional encoding r is calculated as the sum of Rt∈RT×D×1×1, Rw∈R1×D×1×W, Rh∈R1×D×H×1, temporal positional encodings, width, and height depicted as: r=Rt+Rw+Rh. By applying RPE to both Q and K, EFNet can accurately capture the relative positions of frames within a cardiac cycle, ensuring that the model understands how the heart’s structure and motion evolve over time. This spatiotemporal awareness is crucial for the accurate prediction of EF, as it allows the model to account for subtle changes in heart dynamics. The 3D MHSA output of S00US∈RT×D×H×W is represented as:

(4) MHSA(S00US)=concat[softmax(QiKiT+QirKT+rQKiTd)Vi]

where QiKiT is standard dot product of the query and key, QirKT is interaction between the query and the relative positional encoding of the key, and rQKiT represents the interaction between the relative positional encoding of the query and the key. Also, 1d representing a scaling factor for the attention mechanism. T=T12, D=128 (for the second layer), 512 (for the fourth layer), W=W12, H=H12, Qi, Ki, and Vi are queries, keys, and values for the ith attention head derived from 1×1×1 3D convolutions performed over input SUS by WQ(SUS), WK(SUS), and WV(SUS), respectively, and d is D divided by the number of total heads. To estimate the EF, the final RTM’s output is transmitted with one neuron to the fully connected (FC) layer using global average pooling (GAP) (512 input weights). Mean squared error (MSE) is employed as a loss function, while the Adam optimizer is used for model optimization and learning. We performed the final estimation by passing video frames as shown in Fig. 3 in chunks of 16 frames each, after which the output of every segment was averaged for a single patient.

Figure 3 Frames display the blood in the heart’s left ventricle.

Performance and discussion

We discuss the training dataset and present EFNet’s architecture specifics herein section. We contrast EFNet’s performance with that of existing 2D + t spatiotemporal video analysis techniques. We demonstrate the significance of EFNet components that 3D ResNet-18 has included or replaced. We also provide details about the number of trainable parameters of the model and training time.

Implementation

We base our neural network on the 3D ResNet-18 (Tran et al., 2018). The comparison of EFNet’s architectural features with those of 3D ResNet-18 is presented in Table 3. EFNet consists of a three-dimensional convolutional stem, two convolutional stages with a couple of residual modules each, and two stages with two RTMs each. For the purpose of predicting the EF, the outcome of the final RTM is passed with one neuron to the fully linked FC layer after global average pooling (GAP) (512 input weights). We take all the videos, i.e., 10,030, and the training, validation, and test splits are 80%, 10%, and 10%, respectively. We utilized 16 frames and run 45 epochs. We use PyTorch to build our model, then train it on an NVIDIA RTX A-5000 Ti 24-GB GPU along with a mini-batch size of 28, a 0.0001 learning rate at launch, followed by a g = 0.1 step decline after every 16 epochs. We use an ADAM optimizer with a 0.0001 weight decay to reduce the Mean Square Error (MSE) loss function. For each mini-batch, we considered data augmentation during training, such as rotation ( ±25∘), horizontal flip, random brightness, contrast, and blur picture compression. We have used a frame size of 112×112 (original size without any padding or resizing). We have only used frames between ED and ES.

Table 3 Architecture comparison between EFNet and 3D ResNet-18.

For conv3 and conv5, 3×3 3D convolution is swapped by a 3D MHSA.

Stage no	Output result size	3D ResNet-18	EF-Net	
conv1	T0×H0/2×W0/2	3×7×7,64, stride 1×2×2	3×7×7,64, stride 1×2×2	
conv2	T0×H0/2×W0/2	3×3×3,643×3×3,64×2	3×3×3,643×3×3,64×2	
conv3	T0/2×H0/4×W0/4	3×3×3,1283×3×3,128×2	RTM(MHSA)3×3×3,128×2	
conv4	T0/4×H0/8×W0/8	3×3×3,2563×3×3,256×2	3×3×3,2563×3×3,256×2	
conv5	T0/8×H0/16×W0/16	3×3×3,5123×3×3,512×2	RTM(MHSA)3×3×3,512×21×1×1FClayer,GlobalAvgPooling	

Table 4 provides the details of proposed model that combines 3D ResNet-18 with a residual transformer module that uses multi-head self-attention as the attention mechanism. The table identifies the number of trainable parameters, the training time per epoch, the minimum validation mean absolute error (MAE), and the maximum validation R2 score achieved by the model. The minimum validation MAE of 3.92 and the maximum validation R2 score of 0.82 indicates that the model performs relatively well in predicting EF.

Table 4 Model details: 3D ResNet-18 + RTM (MHSA).

Model details	Value	
Trainable parameters	33,147,969	
Training time per epoch (s)	320	
Minimum validation MAE	3.92	
Maximum validation R2 Score	0.81	

Evaluation metrics

Table 5 provides a further information regarding parameter count. Although, EFNet has a higher parameter count compared to some CNN-based methods, but it maintains a balance by investigating the strength of both CNNs and transformers for spatiotemporal analysis. The performance improvements observed in EFNet are due to the innovative integration of RTM within a 3D ResNet backbone, which enables better feature extraction from ultrasound videos. This balance between parameter count and architectural design highlights EFNet’s efficiency, as it outperforms methods with significantly more parameters, such as BERT-based models. Figure 4 representing the validation and training loss, facilitate us to quantify the performance of the regression using root mean square error (RMSE), mean absolute error (MAE), mean square error (MSE), and (R2) in Tables 5 and in 6 respectively.

Table 5 Performance of the proposed EFNet relative to the state-of-the-art methods.

Metrics	Parameter count	RMSE	MAE	R2	
BERT (Reynaud et al., 2021)	346.8 M	7.23	4.90	0.64	
EchoGNN (Mokhtari et al., 2022)	1.7 M	X	4.45	0.76	
EchoGraphs (Thomas & Gilbert, 2022)	27.6 M	5.36	4.01	0.81	
EchoNet (Ouyang et al., 2019)	31.5 M	5.32	4.05	0.81	
EchoCoTr (Muhtaseb & Yaqub, 2022)	X	5.17	3.95	0.82	
3DCNN (Hassan & Obied, 2023)	11 million	1.30	4.71	0.68	
MAFE-Net (Zeng et al., 2023)	X	8.21	6.29	0.54	
EFNet	33 million	4.95	3.79	0.82	

Figure 4 Mean square error (MSE) for training and validation of EFNet using EchoNet dynamic dataset.

Table 6 Performance metrics of the proposed method during training, validation, and testing.

Metrics	MSE	RMSE	MAE	MAPE	R2 score	F1 < 40%	
Train	22.86	4.78	3.64	0.0758	0.84	0.83	
Validation	27.5	5.24	3.92	0.0822	0.81	0.80	
Test	24.5	4.95	3.79	0.0780	0.82	0.81	

Comparison with state-of-the-art algorithms

We performed a comparison of our proposed EFNet with many state-of-the-art 2D + t spatiotemporal video analysis methods. The approach (EchoCoTr) suggested in Muhtaseb & Yaqub (2022) takes on the challenge of calculating the LVEF using ultrasound movies by leveraging the power of vision transformers and CNNs. This method mapped on EchoNet dynamic state-of-the-art dataset with MAE of 3.92 and R2 score of 0.82. In Reynaud et al. (2021), videos of any length are processed and analysis is conducted using a transformer architecture that includes a residual auto-encoder network and BERT model. This study evaluates the EF and yields the MAE of 5.47 and R2 of 0.48, as depicted in Table 5. Echo Graphs Neural Network, proposed in Tran et al. (2018), relies on human intervention to determine critical frames for EF estimation. The echo-Net data set is benchmarked in this study. EchoGNN (Mokhtari et al., 2022) achieved EF prediction with MAE of 4.45 and R2 of 0.76. Mokhtari et al. (2022) makes predictions specifically based on the changes in the left ventricular (LV) during echocardiography films using semi-supervised learning, which are then employed for estimating the left ventricular EF. The specific key points in the left ventricle are learned using graph convolution networks in Dai et al. (2022), which are then used for EF estimation with an MAE of 4.90 and R2 score of 0.79. Table 5 provides a comparative performance summary of EFNet with state-of-the-art contributions. Comparision of the R2 score 0.81 of the proposed approach with respect to the existing methods. The evaluation metrics are encapsulated in Table 5 and confirm the optimum performance of EFNet relative to the state-of-the-art methods.

Figure 5 illustrates the scatter plot of actual vs. predicted EF using the EchoNet Dynamic dataset. The blue line represents the perfect prediction, showing the ideal correlation between the actual and predicted EF values.

Figure 5 Scatter plot between human HF and predicted HF.

Ablation study

Ablation research was conducted to show the effectiveness of recently added components to EFNet. The study involved utilizing a 3D ResNet-18 as the fundamental neural network for the analysis of 2D + t spatiotemporal ultrasound video scans.

In order to acquire a more comprehensive understanding of the cardiac ultrasound video sequence and learn multiple relationships, we have combined convolutional neural network and transformer models. This was achieved by replacing the last and third convolutional layer in the residual module with multi-head self-attention. In addition, we have introduced temporal position encoding (TPE) to enhance the spatiotemporal feature interpretations in both time and space. Through our experimentation, as presented in Table 7, it has been demonstrated that using a combination of CNN and a transformer segment with MHSA and TPE has significantly improved the ability to estimate EF from ultrasound video scans. The selection of 128 frames make sure a balanced temporal resolution for capturing critical phases of cardiac cycle, such as ES and ED. A fixed frame size of 128 was selected to consistently capture at least one complete cardiac cycle, based on typical heart rates and echocardiography frame rates, ensuring accurate identification of key cardiac events while allowing for consistent comparison across datasets. In addition to the experiments with 128 frames, we also conducted trials using 64, 32, and 16 frames to evaluate the impact of frame count on the accuracy of ejection fraction (EF) estimation. These experiments allowed for assessing the trade-off between temporal resolution and computational efficiency, ensuring that even with fewer frames, the key phases of the cardiac cycle (End-Systole and End-Diastole) could still be identified accurately, albeit with reduced temporal detail. This range of experiments demonstrates a thorough exploration of the appropriate frame size for reliable EF calculation.

Table 7 Outcomes of numerous experiments employing CNN and transformer models on the EchoNet-Dynamic Dataset.

The sampling approach used for the BERT, DistilBERT, and EchoNet studies mimics Reynaud et al. (2021), whereas it reflects Muhtaseb & Yaqub (2022) for the EchoCoTr investigations.

Model	No. of frames	Batch size	MAE	RMSE	R2 score	
BERT (Reynaud et al., 2021)	128	2	5.950	8.380	0.52	
Distil BERT (Reynaud et al., 2021)	36	2	6.689	9.234	0.414	
Distil BERT (Reynaud et al., 2021)	128	2	6.430	8.940	0.451	
EchoNet (Ouyang et al., 2019)	36	2	4.05	5.32	0.81	
EchoCoTr-S (Muhtaseb & Yaqub, 2022)	36	25	3.947	5.174	0.82	
MafeNET (Zeng et al., 2023)	X	X	6.29	8.21	0.54	
EchoGraphs (Thomas & Gilbert, 2022)	16	28	4.01	5.36	0.81	
3DCNN (Hassan & Obied, 2023)	X	X	4.71	1.30	0.68	
EFNet (our)	128	2	4.92	7.355	0.59	
EFNet (our)	64	4	4.342	6.046	0.693	
EFNet (our)	36	16	3.92	5.29	0.812	
EFNet (our)	16	28	3.79	4.95	0.82	

Discussion

In this study, we presented EFNet, a technique that extends a 3D ResNet-based network and combines the power of a residual transformer module (RTM) for the interpretation of 2D + t spatiotemporal cardiac ultrasound video data. The outcomes in Table 7 demonstrate that the model trained on just 16 frames with a 3.79 MAE performs better than the most recent models (such as EchoCoTr) on the EchoNet-Dynamic dataset for LVEF prediction (3.92 MAE). Owing to the 2D + t spatiotemporal feature processing, conventional plane identification is not necessary, significantly reducing the workload required to complete the EF estimation. In clinical practice, EFNet may be utilized by doctors as a decision-support tool.

Table 7 makes it abundantly evident that training on 16 frames yields results that are equivalent to those obtained with 128 frames in the BERT model. Nevertheless, none of these experiments performed as well as our proposed method, a result which we attribute to their inability to effectively capture temporal information while recognizing local characteristics in various frames. We share the opinion that EFNet outperformed EchoCoTr-B (Muhtaseb & Yaqub, 2022) and EchoCoTr-S (Muhtaseb & Yaqub, 2022) in our experiments.

In order to acquire a more comprehensive understanding of the cardiac US video sequence by learning multiple relationships, we have combined CNN and transformer models. This was achieved by replacing the last and third convolutional layer in the residual module with multi-head self-attention. In addition, we have introduced temporal position encoding to enhance the spatiotemporal feature interpretation in both time and space. Through our experimentation, as presented in Table 5, it has been demonstrated that with the integration of CNN encapsulated within a transformer-based module, the performance of the EF-estimation test has been greatly enhanced by multi-head self-attention and temporal position encoding.

Although EFNet demonstrates encouraging outcomes, its efficacy is contingent upon the quality and uniformity of the input video frames. Variability in image quality or frame rates, which can differ substantially across ultrasound devices, may influence the precision of the model. Furthermore, the model’s design, which specifies a set number of frames (16), may restrict its utility in contexts where extended sequences are essential to comprehensively capture the dynamic nature of cardiac activities. To ascertain the robustness and generalizability of EFNet, future research should focus on evaluating its performance across a diverse range of clinical environments and ultrasound technologies.

Conclusion

In this article, we introduced the residual transformer module and a 3D ResNet-based network expansion, called EFNet, for analyzing spatiotemporal cardiac ultrasound video scans in 2D + t. Our proposed framework offers a complete approach that can automatically estimate EF without the requirement to recognize standard planes in ultrasound video scans, as is necessary for the traditional technique. We employed a combination of CNN and transformer models for direct EF estimation from ultrasound video scans. The article introduces the residual transformer module which expands a 3D ResNet-based network to analyze 2D + t spatiotemporal cardiac ultrasound video scans. Our system achieves outstanding outcomes, outperforming even top expert physicians using commercial tools, by merging classical and EFNet estimates. In future work, we plan to evaluate EFNet on additional external datasets obtained from various operators and devices with varying levels of experience. We also aim to enhance the model’s performance and robustness by integrating diverse range of data. Our proposed framework has the potential to improve the accuracy and efficiency of EF estimation, making it a valuable tool for clinical practice.

Supplemental Information

Supplemental Information 1 Code and Data.

The authors would like to express their gratitude to the University of Engineering and Technology Health Clinic, whose invaluable contributions and insights have greatly enriched this research. Finally, our appreciation goes to all participants and colleagues who reviewed early versions of the manuscript and provided constructive feedback.

Additional Information and Declarations

Competing Interests

The authors declare that they have no competing interests.

Author Contributions

Waqas Ali conceived and designed the experiments, performed the experiments, performed the computation work, prepared figures and/or tables, authored or reviewed drafts of the article, and approved the final draft.

Wesam Alsabban analyzed the data, prepared figures and/or tables, and approved the final draft.

Muhammad Shahbaz analyzed the data, authored or reviewed drafts of the article, and approved the final draft.

Ali Al-Laith analyzed the data, prepared figures and/or tables, authored or reviewed drafts of the article, and approved the final draft.

Bassam Almogadwy analyzed the data, authored or reviewed drafts of the article, and approved the final draft.

Data Availability

The following information was supplied regarding data availability:

The sample and complete dataset are available at Kaggle: https://www.kaggle.com/datasets/mahnurrahman/echonet-dynamic.

The code and data are available in the Supplemental Files.

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
