# Peer review of "EFNet: estimation of left ventricular ejection fraction from cardiac ultrasound videos using deep learning"

_PeerJ Computer Science, doi:10.7717/peerj-cs.2506_

## Round 0.1 · original submission · Major Revisions

Dear authors,

You are advised to critically respond to all comments point by point when preparing a new version of the manuscript and while preparing for the rebuttal letter. Please address all comments/suggestions provided by reviewers, considering that they should be added to the new version of the manuscript.

Kind regards,
PCoelho

·

Basic reporting

The paper presents EFNet, a novel approach for automated Ejection Fraction (EF) estimation from cardiac ultrasound videos. The authors introduce a Residual Transformer Module (RTM) that extends a 3D ResNet architecture to analyze spatiotemporal (2D + t) ultrasound scans. The method aims to address challenges in manual EF evaluation, such as time consumption and inter-observer variability.
The paper is well-structured and clearly written, providing a comprehensive overview of the problem, the proposed solution, and its implementation. The authors have included relevant citations, demonstrating a good understanding of the current state of the art in automated EF estimation and deep learning approaches in medical imaging.

However, I do have some concerns regarding the methodology section. The description and formulas for the multi-head attention mechanism seem inconsistent with what is depicted in the figures. Specifically, the Relative Positional Encoding (RPE) is shown to be applied to both Q and K in the figures, but this is not reflected in the formulas. Furthermore, the paper lacks a detailed explanation of how the RPE is implemented.

Experimental design

I think that the experimental comparison should include a display of the parameter counts for different methods. This is important because the improvement in performance might simply be due to an increase in the number of parameters rather than an enhancement in network design. How does the parameter count of the method proposed in this paper compare to existing methods? Does it use more or fewer parameters?

Validity of the findings

The design and results of the method proposed in this paper are effectively validated through experiments, demonstrating improved accuracy in EF estimation. However, similar methods and structures may have already been verified in other comparable computer vision video understanding tasks. Therefore, I think the level of novelty in this work is not high.

·

Basic reporting

No comments

Experimental design

The process of identifying the end-systolic (ES) and end-diastolic (ED) frames within a cardiac cycle is crucial for the accuracy of the analysis. However, the manuscript does not provide sufficient detail on how the cardiac cycles were identified within the videos, nor how the specific ES and ED frames were determined. To ensure reproducibility and allow for independent verification of your results, I strongly recommend including a supplemental file that lists the ES and ED frame numbers for each video in your dataset. This file could include a table with columns for the video identifier, the ES frame number, and the ED frame number.
The use of a fixed frame size, such as 128 frames, in Table 5 raises questions regarding its relevance to cardiac cycle analysis. Specifically, if the analysis is centered on identifying specific points like end-systolic (ES) and end-diastolic (ED) frames, it would be beneficial to understand:
1. Why 128 Frames?
o What is the rationale for choosing a fixed frame size of 128?
o Are these frames intended to cover multiple cardiac cycles, or is there a specific sampling strategy used to select these frames?
2. Relation to ES and ED Frames:
o How does the fixed frame size of 128 relate to the identification of ES and ED frames?
o Are these specific frames within the 128-frame sequence, or is the analysis focusing on the overall temporal dynamics of the heart?

The techniques mentioned in the Dataset section (rotation, flipping, brightness/contrast adjustment, compression, and blurring) are commonly referred to as data augmentation techniques rather than pre-processing techniques. Data augmentation is used to artificially expand the training dataset by applying various transformations to the images, which helps improve the model's generalization.
The manuscript does not specify how many frames were considered for each video or how the varying frame rates in the dataset were handled. Given that different frame rates can significantly affect the analysis of cardiac cycles, it is important to clarify how these variations were addressed. Did you normalize the frame rates across videos or apply any specific technique to ensure consistency in the analysis?

For the Selection of Frame Segments it’s not entirely clear how the specific 16-frame segments are selected for input. If different videos have different lengths or frame rates, how are the frames chosen to ensure consistent representation across videos? Are all possible 16-frame segments used, or is there a sampling strategy involved?

The manuscript specially the figure 2. does not provide a clear description of the output layer's structure and how it directly relates to the calculation of ejection fraction (EF). It's important to understand:
• How does the network convert the processed features from the final convolutional layers into a specific EF value?
• Is there a specific regression layer at the output, such as a fully connected layer followed by a linear activation function, that produces a continuous value representing the EF?
• How does the network ensure that the predicted EF is within a realistic range (e.g., 0% to 100%)?
Table 5 presents results for models considering inputs with frame numbers of 64 and 128, but there are many videos in the dataset with lengths shorter than 64 frames. This raises a concern about how the model handles videos with fewer frames than the required input size.

The paper does not include a scatter plot with a regression line comparing the model-predicted ejection fraction to the human-calculated ejection fraction. Including such a plot would be valuable for visualizing the correlation and relationship between these two sets of measurements.

Validity of the findings

No comments

Additional comments

No comments

·

Basic reporting

The writing is clear. The introduction presents the background including reference to relevant previous work.

Experimental design

The research question is clearly defined and the method is presented in detail with relevant figures.

Validity of the findings

Conclusions are not fully supported by the results, see additional comments.

Additional comments

Specific comments
• Line 54: “and a better prognosis may be achieved (36)”
o I suggest you provide a medical reference to this statement, or consider removing it
• Line 55: Normal range for EF is not accurate. Please see e.g., Lang et al. Recommendations for Cardiac Chamber Quantification by Echocardiography in Adults. J Am Soc Echocardiogr 2015;28:1-39
• Line 58: I guess reproducibility might be a better term here
• Line 120: “Shi et al”
o To what does this refer?
• Lines 142-144: Is this an introductory statement or a conclusion?
• You refer to the Echonet-Dynamic dataset of 10,030 recordings.
o Still, it would be benefitial if you could provide a brief overview of the constitution of this dataset (age, sex, BMI, diseases, EF, image quality etc), which is of importance when considering the generalizability of your results
o You provide data on 7 training cases, 2 validation cases and 1 test case in the supplementary material. This is confusing. Please explain.
• How did you define/detect ED and ES? Manually assigned? Autodetection?
• EF definition is repeated, which ideally should be avoided
• Good results with a lower number of frames from each case/recording could be a significant finding, as it would potentially lead to more resource effective training. However, at the same time you have altered the batch size. How can you discriminate the effects of these interventions? What are the effects if you change the number of frames without altering the batch size?
• What is the feasibility of your method?
• Lines 277-278: “EFNet 278 outperformed EchoCoTr-B(20) and EchoCoTr-S (20) in our experiments”
o I would argue that this is slightly overstated. Is this difference clinically meaningful?
• Lines 300-301: “outperforming even top expert physicians using commercial tools”
o I cannot see how your results support this statement

---

## Round 0.2 · accepted · Accept

Dear authors, we are pleased to verify that you meet the reviewer's valuable feedback to improve your research.

Thank you for considering PeerJ Computer Science and submitting your work.

Kind regards,
PCoelho

·

Basic reporting

The paper presents EFNet, a novel approach for automated Ejection Fraction (EF) estimation from cardiac ultrasound videos. The authors introduce a Residual Transformer Module (RTM) that extends a 3D ResNet architecture to analyze spatiotemporal (2D + t) ultrasound scans. The method aims to address challenges in manual EF evaluation, such as time consumption and inter-observer variability.
The paper is well-structured and clearly written, providing a comprehensive overview of the problem, the proposed solution, and its implementation. The authors have included relevant citations, demonstrating a good understanding of the current state of the art in automated EF estimation and deep learning approaches in medical imaging.

The revised version has addressed my concern and corrected the formulation of the multi-head attention.

Experimental design

The revised version addressed my concern and provided the number of parameters in EFNet, which has comparable parameters and achieves SOTA performance.

Validity of the findings

The design and results of the method proposed in this paper are effectively validated through experiments, demonstrating improved accuracy in EF estimation. However, similar methods and structures may have already been verified in other comparable computer vision video understanding tasks. Therefore, I think the level of novelty in this work is not high.

·

Basic reporting

no comment

Experimental design

no comment

Validity of the findings

no comment

Additional comments

The authors have provided throrough replies and revisions to manuscript. I only have to minor comments:

- Line 56 in tracked changes document: "in females" missing?

- I previously asked how you could discriminate the effects of reduced reduced frame rate and increased batch size when both interventions are performed simultaneously. In response, you state: "Therefore, controlling one variable at a time during experiments is crucial to draw valid conclusions regarding their individual impacts on model performance and resource efficiency."
Hence, you argue against your approach, which alters the batch size and no of frames at the same time. Does this warrant a comment/discussion in the manuscript?